# Effect of a Simulated Activity on Student Ability, Preparedness, and Confidence in Applying the Pharmacists’ Patient Care Process to Contraceptive Prescribing

**DOI:** 10.3390/pharmacy8030146

**Published:** 2020-08-17

**Authors:** Sarah E. Lynch, Brooke Griffin, Autumn Stewart-Lynch, Kathleen M. Vest

**Affiliations:** 1School of Pharmacy and Pharmaceutical Sciences, Binghamton University, Binghamton, NY 13902, USA; 2Chicago College of Pharmacy, Midwestern University, Downers Grove, IL 60515, USA; bgriff@midwestern.edu (B.G.); kvestx@midwestern.edu (K.M.V.); 3School of Pharmacy Duquesne University School of Pharmacy, Pittsburgh, PA 15282, USA; stewar14@duq.edu

**Keywords:** hormonal contraception, contraception, birth control, pharmacists’ patient care process, pharmacist prescribing, pharmacist access, PPCP

## Abstract

Several states now permit pharmacists to prescribe hormonal contraception. Consequently, some schools of pharmacy now incorporate activities intending to prepare students to offer this service. This study aimed to assess the impact of a simulated activity on student pharmacists’ readiness for, ability to use, and confidence in applying the Pharmacists Patient Care Process along with the United States Medical Eligibility Criteria to a patient seeking contraception. Students completed a contraceptive-prescribing simulation with standardized patients. Scores were analyzed for safe and appropriate prescribing practices. Pre- and post-workshop surveys measured confidence and perceived preparedness. Chi-square and Mann–Whitney U tests were used to analyze categorical variables and Likert-scale data, respectively.The mean activity score was 86% (median 90%), with significant change in student confidence of ability to complete the process (*p* < 0.0001). The majority of students at baseline (52.2%) and follow up (53.2%) reported needing more practice during advanced pharmacy practice experiences (APPEs) to feel prepared. There was a significant change pre/post in students who agreed that their curriculum prepared them (15% to 28.7%, *p* = 0.0014). This study suggests that students are able to safely and appropriately prescribe contraception in a simulated activity. The activity increased student reported confidence and moved some students towards readiness for contraceptive prescribing.

## 1. Introduction

Schools of Pharmacy are required by the accrediting body, the Accreditation Council for Pharmacy Education (ACPE), to ensure that graduates are “practice-ready”, defined as “able to provide direct patient care in a variety of healthcare settings” [1]. The “Pharmacotherapy Didactic Curriculum Toolkit” is a guidance document published by the American College of Clinical Pharmacy Educational Affairs Committee for schools of pharmacy regarding the development, maintenance, and revision of pharmacy curricula in consideration of practice changes and advances in the field. In the 2016 toolkit, the therapeutic area of hormonal contraception is identified as a practice competency Tier 1, further defined as one in which: “Students receive education and training on this topic to prepare them to provide collaborative, patient-centered care on graduation and licensure” [2]. Federal and state regulations require pharmacists to protect the health and safety of the public in whom they have been entrusted. Furthermore, to be considered “practice ready”, student pharmacists will need to be prepared for the responsibilities of initiating, adjusting and managing therapy for many common acute and chronic disease states. Proposed provider status laws, including one seeking federal recognition as providers under Medicare, may lead to additional public expectations about pharmacist scope and ability [3]. Pharmacists’ prescriptive authority in the United States falls under these four categories: patient-specific collaborative prescribing through collaborative practice agreements (CPAs), population-specific prescribing through CPAs, statewide protocols, and class-specific prescribing [4]. As of June 2020, eleven states have regulations that specifically permit pharmacists to prescribe or furnish hormonal contraceptive therapy and this number is expected to increase with additional legislation proposed in several other states [5]. Oregon was the first to develop a statewide protocol to allow prescribing of pharmacist contraceptives. In a 2019 cost-effectiveness analysis of pharmacist contraceptive prescribing in Oregon, Rodriguez et al. demonstrated cost savings of $1.6 million dollars by preventing 51 unintended pregnancies [6].

State legislation varies in the training requirements of pharmacists to become eligible to prescribe hormonal contraception. For example, one state requires graduation from a school of pharmacy located in the state, while other states require multi-hour training programs. In some states, the training programs are being incorporated in the required pharmacy school curricula within the state and/or provided through continuing education. However, ACPE does not currently dictate how much or what type of training student pharmacists should receive to prepare them for comprehensive hormonal contraceptive prescribing. It is not currently known what is the appropriate amount of training required for pharmacists to be able to safely, effectively, and confidently prescribe hormonal contraceptive therapy to patients using the Pharmacist’s Patient Care Process (PPCP), the standard approach to patient care used by pharmacists in all settings [7]. Therefore, it is inferred that data are needed to determine whether the traditional teaching methods typically employed in schools of pharmacy are adequate to ensure student pharmacists are indeed “practice ready”, as this activity becomes an expectation in standard practice for many pharmacists.

We hypothesize that mostly didactic instructional methods and current length of curricular delivery may be inadequate to prepare student pharmacists to safely and appropriately prescribe hormonal contraceptives. We also anticipate that student pharmacists will be significantly more confident in their ability to prescribe hormonal contraceptives after completing a hormonal contraceptive prescribing simulation as part of skills-based lab exercise. 

## 2. Materials and Methods 

This was a quasi-experimental study of student ability to effectively and confidently prescribe contraceptives per a pharmacist protocol during a pharmacy practice simulation lab activity. Student pharmacists at two colleges of pharmacy located in states where pharmacists do not currently prescribe hormonal contraceptives were recruited to participate in the study. At both institutions, students learn this topic in the Fall of their second professional year. Content delivery and length varies in each program. Each program has a workshop or skills lab that is utilized for students to apply the concepts learned during the required didactic therapeutics module. Assignment scores were used to determine student efficacy and ability, and a survey was administered to determine student confidence and opinions about preparedness and training. 

The students first participated in the hormonal contraceptive unit within their school’s required pharmacotherapeutics sequence. The faculty teaching these courses taught the materials according to their standard didactic methods, meaning that materials were not covered any differently to how they had been covered in previous course deliveries that were not done in conjunction with this lab activity. For both institutions, this meant that material was primarily delivered via case-based lecture, accompanied by optional readings, homework, and student self-study.

After the lecture-based material and prior to the lab, students independently reviewed a new 20-min voice-over power-point with information about pharmacist-prescribed hormonal contraception. This presentation described new state regulations that allow pharmacists to prescribe contraceptives. It also covered the process of prescribing, as well as Oregon’s original pharmacist-prescribing protocol, and related these steps to the Pharmacists’ Patient Care Process (PPCP) steps (collect-assess-plan-implement-follow-up/monitor). 

An electronic survey was developed and administered via Qualtrics® (Provo, UT, USA) to students twice: immediately prior to and again after completing the lab-based activity. Students were provided with assigned codes by a research assistant; these codes were kept anonymous from the researchers and were used to match student post-survey opinions and score on the activity. Survey completion was optional and was not a graded component of the lab activity. Students were given the option to have their scores included in the final analysis. The survey included questions about demographics, previous practice, and self-reported time spent learning and studying the therapeutic material. There were also two categories of questions for students to rate their opinions based on a Likert scale. These two categories were confidence in performing various steps of contraceptive prescribing, which were mapped to the PPCP, and their perception of preparedness to prescribe. 

For the lab-based activity, students were required to collect and analyze patient information in order to make a recommendation to an individual seeking a pharmacist to prescribe them hormonal contraception. This activity was newly designed to replicate realistic hormonal contraceptive services being offered by pharmacists. Researchers developed multiple realistic patient scenarios about women seeking to have a hormonal contraceptive prescribed by a pharmacist. Each scenario had a “challenge” element, which could have included medical conditions, drug interactions, patient preferences, or past failure of a product. Pharmacy students were provided with a partially completed patient intake form and were then given time to interview their patient for additional information. The patients were played by standardized patients (SP), either a paid actor or an upperclassman student, depending on the college. These individuals were provided with the scenario prior to the lab in order to know their role and to be able to answer any questions the students may pose. The students interviewed their SP as a group of five to ten students (depending on the institution), then had to individually complete a post-activity assignment. One school utilized simulated clinic rooms for patient and student meetings, and the other school’s session took place in normal classrooms, but students were given separate areas to carry out their question and answer session with the SP. The assignment consisted of four clinical questions specific to the assigned patient case, and included providing a recommendation and rationale. Students were told that their “formulary” included any oral hormonal contraceptive, the vaginal ring, the transdermal patch, the subcutaneous injection, or the option to refer. There were also four PPCP-based questions related to the pre-lab activity about the general process of pharmacist prescribing. After the assignment was turned in, students completed the post-survey. 

The post-assignment was graded and the post-survey results were matched to the scores using the generated codes for students who authorized this. The research assistants matched the post-assignment grade to the survey result and shared it with the researchers so that the student identity would not be known by the researchers/faculty members. 

Survey responses were analyzed in aggregate. Survey responses were included even if the individual did not complete the entire survey. Data from Likert confidence scales were converted from ordinal to dichotomous variables with “Strongly Agree” and “Agree” categorized as “Confident” and “Strongly Disagree” and “Disagree” categorized as “Not Confident”. The Chi-square test was used to analyze categorical data related to factors affecting confidence and self-perceived preparedness. Pre- and post-survey results were compared using the Mann–Whitney U test to determine if there was a significant change in confidence level after completing the simulation activity. Spearman’s Rho was used to assess for correlations between knowledge (as measured by scores on the activity) and confidence, knowledge, and time spent engaged with learning material, and knowledge and frequency of past experience counseling patients.

## 3. Results

### 3.1. Pre-Survey

#### 3.1.1. Baseline Demographics

A total of 216 responses were received for a 91.1% response rate. The mean age was 24.5 years (range 20–45, SD 3.8) (199 responses) and 127 (62%) participants were female (Table 1). The majority of respondents (41.2%) were Caucasian, followed by Asian/Pacific Islanders (38.7%). Of these respondents, 193 (69.2%) reported previous experience in community/retail pharmacies and 50 (17.9%) had experience in hospital/institutional pharmacies. Of those responding, 61.3% had not counseled a patient on the use of a contraceptive product in the previous 6 months; 21.7% had “1–3 times”. Enrollment in the second professional year was selected by 99% of respondents. Table 2 summarizes student self-reported estimates of time spent in didactic instruction, studying for the course, or for personal interest. 

#### 3.1.2. Confidence

As seen in Table 3, students at baseline were the most confident in their ability to gather information (PPCP—collect) from a patient seeking contraception (123, 59.1%, strongly agreed or agreed) and were the least confident in their ability to recommend (PPCP—plan) a specific hormonal contraceptive product (83, 40.3%, strongly agreed or agreed). Students reporting having spent four to six hours in didactic training or outside of class were more confident than those spending less time: 71.2% vs. 53.8% (*p* < 0.031228) and 74.2% vs. 45.5%, 56.8% (*p* < 0.4656), respectively. Spending four to six hours outside of class also resulted in significant differences in confidence related to collecting information (*p* < 0.031434) and monitoring therapy (*p* < 0.020296) compared to those spending less time. Increasing amounts of time spent in self-study also resulted in more confidence in collecting information, assessing patients, choosing a specific products, and providing patient counseling. There were no differences in confidence between male and female respondents. 

#### 3.1.3. Preparedness

Prior to the simulation exercise, thirty-one (14.98%) students agreed to feeling prepared based on the training received through their curriculum, however, 108 (52.17%) anticipated that practice during APPEs would provide the needed preparation.

### 3.2. Simulation Activity

In aggregate, the activity was completed by 228 students. The mean score on the workshop activity was 86%; the median was 90%. Fifty-four students (23.7%) scored <80% on the activity. Sixty-seven students consented to having their activity scores included in the analysis with survey data and provided a correct unique code to allow for correlations between knowledge and other variables. Weak correlations (r < 0.2) existed between simulation activity scores and time spent learning material and between knowledge and frequency of previous counseling experiences. 

### 3.3. Post Survey

#### 3.3.1. Demographics

A total of 100 responses to the follow-up survey were received and of these there were 94 usable responses. The mean age was 24.6 years (range 20–45, SD 4.12) and 62 (66.0%) were female (Table 1). The majority of respondents (41.5%) were Asian/Pacific Islanders followed by (40.4%) Caucasian. Of these respondents, 61 (64.9%) reported previous experience in community/retail pharmacies and 23 (24.5%) had experience in hospital/institutional pharmacies.

#### 3.3.2. Confidence

There was a significant change in pre- and post-student confidence of their ability to complete each step of the PPCP (*p* < 0.0001) with less than 60% of students expressing confidence in all steps at baseline (collect = 59%, assess = 48%, plan = 52%, implement = 57%, and follow-up/monitor = 52%), and over 80% expressing confidence at follow-up (91%, 87%, 88%, 87%, 82%) (Table 2). Weak correlations (r < 0.2) existed between simulation activity scores and confidence in each step of the PPCP. Table 2 summarizes student-reported confidence in PPCP abilities before and after the simulation activity. 

#### 3.3.3. Preparedness

The majority of students at follow up (53.2%) reported a need for more practice during APPEs to feel prepared, which was unchanged from baseline. However, there was a significant change (91.3% increase) in the number of students in agreement with the statement “I feel prepared based on the training I have received through my school’s curriculum”. At baseline, 15% were in agreement compared to 28.7% after completion of the workshop (*p* = 0.00614). Fewer students felt the need for additional training from an employer (3.2%) or through a continuing education program (2.1%) at follow-up. Of the 17 students scoring <80% on the activity, 6 (35%) stated “I feel prepared based on the training I have received through my school’s curriculum.” A summary of the responses at baseline and follow-up related to self-perceived preparedness are provided in Table 4.

## 4. Discussion

To our knowledge, this is the first study of its kind to evaluate (1) how students apply the PPCP to contraceptive services and (2) an identical contraceptive services assessment at two different colleges of pharmacy. 

### 4.1. Pharmacists’ Patient Care Process

As colleges of pharmacy are introducing and implementing the PPCP in their curriculum, it is important to include and assess this process within innovative practice models and community-based services, such as contraceptive prescribing. In this study, students’ baseline level of confidence was lowest at the “implement” step of the PPCP related to this therapeutic topic. However, after the opportunity to practice this activity, confidence in every level of the PPCP significantly improved. This suggests that direct application of a simulated experience may help reinforce aspects of the PPCP. As new practice models emerge for pharmacists, all areas of the PPCP will need to be well understood for practice transformation to occur. 

Although not a direct objective, Landau et al.’s 2009 survey of practicing pharmacists used elements of the PPCP when asking about comfort level. Results of that study demonstrated that the majority of respondents were comfortable with all of the following responsibilities: asking questions about risk assessment and sexual history (collect), measuring blood pressure (assess), educating on proper use (implement), and scheduling follow-up (follow-up) [8]. Interestingly, students in our study reported the highest confidence in the process areas of collect and implement, which align with more traditional pharmacist roles of contraceptive dispensing and counseling. This was a similar finding to a study of the PPCP’s integration into a disease management course where students demonstrated the most understanding of the lowest levels: collect and implement [9]. In another study of PPCP integration in an integrated pharmacotherapy course, the authors concluded that although application exercises embedded into courses take additional faculty time for development and execution, these activities met their desired outcomes and are highly valuable [10]. In addition, the author of that study believes that students would benefit from more time with practicing aspects of the PPCP, as allowing students time and space to “think like a pharmacist” in their assessment and plan is an important part of the critical thinking process [10].

As more states allow pharmacists to prescribe contraception under statewide protocols, evaluating contraceptive eligibility (assess), recognizing when a referral is necessary (plan), and recommending a contraceptive product (plan) are critical skills needed for contraceptive patient care. 

### 4.2. Curricular Exposure to Contraception

Realizing that college-aged students are likely to learn about contraception in their personal lives, we sought to quantify the total amount of time they spent learning and how it may impact their confidence and academic performance. Results suggest that the amount of time students spend learning and studying contraception, both inside and outside of the classroom, may be an important factor in student confidence in the process of providing contraceptive services. Confidence was significantly higher in students who spent additional time learning about contraception outside of class. Students who performed more self-study of contraception demonstrated greater confidence in several aspects of the PPCP. Greater confidence observed in groups with time spent in didactic study between four and six hours may provide guidance for curricular decisions around the time allocated for these topics to facilitate adoption of these practices by new pharmacists. Colleges of pharmacy can consider a minimum number of hours for activities related to contraceptive services. Surprisingly, time spent learning was not correlated with performance scores on the activity. However, the generally high scores on the activity itself and the small number of students providing activity score data may have contributed to these comparisons lacking sensitivity and power to detect differences. 

### 4.3. Preparation and Ability

Although a majority of students reported feeling unprepared prior to the simulation activity, results suggest that the activity moved some students towards readiness for contraceptive service provision. Scores on this activity demonstrated that students are able to safely and appropriately prescribe contraception in a simulated activity. The percentage of students who felt prepared based on the school’s curricular coverage of the topic significantly improved and almost doubled after completion of the simulation activity. This suggests that didactic learning alone may not suffice; additional hands-on training, such as a simulation workshop, helps students feel more prepared. This is reflected in the general practice environment as well. In one study of community pharmacists who were surveyed before and after a contraceptive service training program, more pharmacists reported a significantly higher level of comfort after the training [11].

Our study and previous studies have not only exposed students to the Center for Disease Control’s Medical Eligibility Criteria (CDC MEC), a critical guideline document, but have also provided an opportunity for students to practice using it [12]. Although not measured in our study, a previous study demonstrated that roughly 25% of practicing pharmacists surveyed were not familiar with the CDC MEC [13]. Interestingly, one-third of those respondents had graduated within the previous five years, and three-quarters of those respondents were graduates of pharmacy schools in California, where no additional training is required for graduates after 2014. These findings may indicate that a gap may exist in current pharmacy curricula related to preparedness for contraceptive prescribing services [13]. Since this tool is required in the provision of contraceptive services, it is possible that practicing pharmacists may seek additional training because they are not aware that this eligibility tool exists or they have not had the opportunity to apply its utility to patient cases. A lack of knowledge of the CDC MEC may also negatively impact pharmacist perception of their ability or interest in providing contraceptive prescribing services [13]. Early exposure as students to the tools and procedures required to carry out this process could impact future interest in offering services in this area that currently may be limited by confidence or perceptions of preparedness.

### 4.4. Confidence

The lab activity increased student-reported confidence in their ability to complete each step of the PPCP as it relates to contraceptive services. Following the contraceptive recommendation activity, students’ self-perceived confidence significantly improved compared to what it was before the activity. In a small 2018 study (*n* = 11), similar high confidence scores among students were demonstrated, with a simulated contraceptive recommendation activity when the students were surveyed after completing the activity [12]. 

Interestingly, although student performance was generally strong, over one-third with low scores demonstrated a high level of self-perceived confidence, despite poor academic performance. This may suggest that self-perceived confidence surveys may not be enough to assess a student’s readiness, as some students’ scores reflect an overconfidence based on their performance. 

### 4.5. Student Opinions of Training Needs

The number of students who agreed that the school’s curriculum prepared them changed greatly from before to after the activity, however, even after the simulated activity, this percentage was still less than one-third of students. This is less than what was reported in a 2017 study of pharmacists; practicing pharmacists were asked whether pharmacists were prepared for providing contraceptive services, with 50% stating yes [14]. 

When asked in which areas students would like to receive additional training, there was general agreement that additional training is needed. This was similar to the results demonstrated in surveys of practicing pharmacists and students [8,13,14,15,16]. In our study, approximately half of the students felt they would need more practice during APPEs to feel prepared (both before and after the activity). This differs from previous studies of practicing pharmacists who preferred additional training through continuing education programs, live seminars, and written materials [8,13,14]. Surveys of practicing pharmacists would naturally not list APPEs as a training option, as in our study. However, it highlights that pharmacy students may believe they will obtain training to provide contraceptive services during their experiential rotations, which would be challenging for most colleges to provide, especially those where communities are not currently allowed to offer contraceptive services. Women’s health rotations are not required, and most ambulatory care and community rotations do not include women’s health activities. This suggests a gap between the expectations of current educational training and future responsibilities in practice. This insight should encourage colleges of pharmacy to include more training to prepare students for IPPEs and APPEs, as additional learning may be limited during experiential learning. 

Only a small percentage of students agreed that other types of training would help them feel prepared. On the pre-survey, a very small percentage of students (~10%) indicated that additional training through their employer or continuing education programs would be needed. Even fewer agreed with this on the post-survey, with only ~3% of students indicating that same need. With this data, colleges of pharmacy hold the responsibility to ensure adequate training prior to graduation.

Approximately 10% of students, both pre- and post-survey, indicated that advanced training or certificate programming would be needed. This is surprising given that most of the training that currently occurs is through state association and board-approved training programs. 

Traditionally, pharmacy students have been taught how to determine eligibility for contraceptive products. Herein, they are already prepared for collecting and assessing therapeutic information. What is lacking is educational efforts in the implementation and follow up for the innovative service, which also includes the operational aspects of prescribing and billing. In a recent commentary, authors recommended a consistent training program for pharmacist provision of contraception [17]. The authors supported a consistent training program in order to facilitate credentialing, to provide a systematic approach to care, and to gain support from employers, payors, and patients [17].

### 4.6. Limitations

This study researched an intervention at two colleges of pharmacy, so the results may not reflect the opinions and hours of study of all pharmacy students nationally. Both of the colleges in this study taught contraception during the second year of the curriculum, so some results, such as self-perceived confidence, may be biased (Dunning–Kruger Effect). The results of this study represent one moment in time, whereas confidence in providing contraceptive services may increase with post-graduate training and/or practice or decrease with length of time from learning the material.

Although the ACCP Pharmacotherapy Didactic Toolkit identifies contraception as a recommended Tier 1 topic for practice competency, professional organizations and accrediting bodies do not dictate how many hours should be dedicated to this topic [2]. The results of this study suggest that colleges of pharmacy consider a minimum of four to six hours of didactic coursework with simulation activities to be a benchmark for student preparedness.

With the emergence of legislation allowing pharmacists to prescribe contraception in certain states, it has been identified that there is little standardization in contraceptive curriculum across colleges of pharmacy [18,19]. With this variation in mind, it is possible that some pharmacy students are graduating with an advantage in this new practice area. It is also notable that states require varying training opportunities for pharmacists to prescribe contraception [18]. It is possible that inconsistency in curricular training and inconsistency in state-mandated training will lead to practice inconsistencies. In an effort to promote successful implementation of this expanded professional service, colleges of pharmacy should aim for similar levels of preparedness before graduation (Table 5).

## 5. Conclusions

Students at two colleges of pharmacy generally demonstrated competence with a simulated contraceptive recommendation service activity. After the activity, students reported an increase in self-perceived confidence in this therapeutic process compared to before the activity. Not all areas of the PPCP were associated with high levels of confidence, which may help colleges of pharmacy plan for PPCP application-based activities. Targeted training above and beyond typical pharmacy school curricula may be required to give pharmacists the confidence they need to feel adequately prepared. Additional research is needed with more colleges of pharmacy to determine student knowledge, student competence, and which pedagogical methods are most effective (i.e., didactic, independent study, simulation) for educating students on pharmacist contraceptive services.

## Figures and Tables

**Table 1 pharmacy-08-00146-t001:** Participant Demographics at Baseline and Follow-Up.

	Pre-Survey (*n* = 216)	Post-Survey (*n* = 94)
Mean age (in years), SD *n* = 199	24.5, 3.8	24.6, 4.12
Female	127 (62%)	62 (66.0%)
Ethnicity		
Caucasian	84 (41.2%)	38 (40.4%)
Asian/Pacific Islander	79 (38.7%)	39 (41.5%)
Hispanic or Latino	11 (5.39%)	6 (6.4%)
Black or African American	10 (4.9%)	5 (5.3%)
Native American or American Indian	0 (0%)	0 (0%)
Other	20 (9.8%)	6 (6.4%)
PY2 (second professional year)	203 (99%)	93 (98.9%)
Previous Work Experience (participants could select more than one)		
Community/Retail Pharmacy	193 (89.4%)	85 (90.4%)
Hospital/Institutional Pharmacy	50 (23.2%)	24 (25.5%)
Long term Care	9 (4.2%)	5 (5.3%)
Specialty Pharmacy	13 (6.0%)	1 (1.1%)
Managed Care	4 (1.9%)	2 (2.1%)
Other	10 (4.6%)	3 (3.2%)

**Table 2 pharmacy-08-00146-t002:** Student-Reported Prior Instruction and Study of Contraception.

Prompt	0 h	1–3 h	4–6 h	>6 h
As best as you can recall, how many hours would you estimate that you have spent learning about contraception in didactic instruction (e.g., in-class, lecture, etc)? (*n* = 213)	18 (8.45%)	132 (61.97%)	52 (24.41%)	11 (5.16%)
As best as you can recall, how many hours would you estimate that you have spent learning about contraception out of class (e.g., studying for an exam, preparing for lab, etc)? (*n* = 213)	44 (20.66%)	118 (55.4%)	31 (14.55%)	20 (9.39%)
As best as you can recall, how many hours would you estimate that you have spent learning about contraception through self-study (e.g., individual interest, personal research, etc)? (*n* = 212)	74 (34.91%)	107 (50.47%)	22 (10.38%)	9 (4.25%)

**Table 3 pharmacy-08-00146-t003:** Students’ Reported Confidence in Pharmacist Patient Care Process (PPCP) Abilities Related to Pharmacist-provided Contraception Before and After Simulation.

Survey Item	Pre (*n* = 207)/Post (*n* = 94)	Strongly Agree (5)	Agree (4)	Not Sure (3)	Disagree (2)	Strongly Disagree (1)	Median (Average)	Z-Score (*p*-Value)
I am confident in my ability to gather information from a patient seeking contraception (COLLECT)	Pre *	15.38%	43.75%	25.96%	12.98%	1.92%	4 (3.6)	−5.51998 (<0.00001)
	Post	34.04%	57.45%	8.51%	0.00%	0.00%	4 (4.26)	
I am confident in my ability to evaluate a patient seeking contraception (ASSESS)	Pre	11.11%	36.71%	34.30%	15.46%	2.42%	3 (3.4)	6.21338 (<0.00001)
	Post	27.66%	59.57%	11.70%	1.06%	0.00%	4 (4.14)	
I am confident in my ability to recognize when a patient needs to be referred to another healthcare provider for contraception (PLAN)	Pre	13.53%	38.65%	36.71%	8.70%	2.42%	4 (3.5)	−5.85684 (<0.00001)
	Post	31.91%	56.38%	11.70%	0.00%	0.00%	4 (4.20)	
I am confident in my ability to recommend a specific hormonal contraceptive product (PLAN)	Pre	9.71%	30.58%	32.04%	21.36%	6.31%	3 (3.2)	−6.04292 (<0.00001)
	Post	22.34%	57.45%	14.89%	5.32%	0.00%	4 (3.97)	
I am confident in my ability to provide counseling on contraceptive products (IMPLEMENT)	Pre	16.59%	40.00%	26.34%	14.15%	2.93%	4 (3.5)	−4.60104 (<0.00001)
	Post	27.66%	59.57%	10.64%	2.13%	0.00%	4 (4.13)	
I am confident in my ability to identify appropriate monitoring parameters for a patient using contraceptive products (MONITOR/FOLLOWUP)	Pre	12.62%	39.32%	28.16%	16.50%	3.40%	4 (3.4)	−4.9376 (<0.00001)
	Post	25.53%	56.38%	15.96%	2.13%	0.00%	4 (4.05)	

* *n* = 208.

**Table 4 pharmacy-08-00146-t004:** Students’ Reported Preparedness for Prescribing Hormonal Contraceptives Before and After Simulation.

	Before Activity (*n* = 197)		After Activity (*n* = 92)		
	No. of Respondents	Percentage of Respondents	No. of Respondents	Percentage of Respondents	% Change from Baseline
I would need more practice during APPEs to feel prepared.	108	52.2%	50	53.2%	+1.9%
I would need more training through my employer to feel prepared.	22	10.6%	3	3.2%	−69.8%
I would need more training through an advanced training or certificate program to feel prepared.	19	9.2%	10	10.6%	+15.2%
I feel prepared based on the training I have received through my school’s curriculum.	31	15.0%	27	28.7%	+91.3%
I would need more training through continuing education programs.	17	8.2%	2	2.1%	−74.4%
I would need more practice in a PGY1 residency program to feel prepared.	7	3.4%	2	2.1%	−38.2%
Other	3	1.4%	0	0.0%	−100.0%

**Table 5 pharmacy-08-00146-t005:** Call to Action.

Further Research is Recommended to Examine:
Whether knowledge obtained through traditional didactic approaches is adequate to prepare students for this advanced role. Whether incorporation of simulated experiences as part of the didactic curriculum influences student perceptions of confidence. The role of contraceptive experience during APPEs in student confidence and knowledge.

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
