# Peer review of "Effect of a Simulated Activity on Student Ability, Preparedness, and Confidence in Applying the Pharmacists’ Patient Care Process to Contraceptive Prescribing"

_pharmacy, 2020, doi:10.3390/pharmacy8030146_

Round 1

Reviewer 1 Report

General Comments

This is an extremely well written and organized paper. The authors present a very interesting educational study about students’ confidence and abilities in providing hormonal contraceptive services after going through a curricular training. This paper adds to the existing literature. Thank you!

Throughout the paper, indicating the training is for “prescribing” such as Line 245 is misleading. This may be a terminology difference, but what we are really training students and pharmacist is to provide hormonal contraception services, from which a prescription may result; a key component, as authors identify is the Plan step (actually making a decision on which hormonal contraception to choose/recommend). Would be helpful to add some language in the paper that students are being prepped to provide HC services (not just prescribing a drug).

Specific Comments

Line 64: Do authors mean pharmacy students or pharmacists (as written)? Please clarify.

Line 68: Suggest rewording that two examples of schools…As worded, makes it sound like only 2 schools have this content.

Line 86: More detail on what the “module” consists of would be helpful. In Line 91, authors indicate “lecture-based” material—I assume this is the module? How long is the lecture-based material? Was there any assigned reading or other material to review as self-study?

Line 91: It isn’t clear what was new in the training, if anything. Was the pre-recorded 20 minute PPT a new component to the training? The authors’ hypothesis is that the current length of curricular content may be inadequate, which initially led me to think the study was going to assess 2 different types of training, but this isn’t the case. It appears that the addition of the simulation component was a new piece.

Line 109: Suggest use of term “seeking” instead of “looking”.

Line 116: How large were the student groups? Did the interview with the SP occur in a Simulation Center? Or other type of room? Please clarify.

Lines 152-153: Where is this data presented? How many students indicated 4-6 hours? Suggest including the mean/median of self-study hours (as this is also cited on Line 173).

Lines 184-185: I am confused why the %’s are negative if confidence increased post-training/simulation. Please clarify.

Line 265: Not quite sure the actual implications here. What year were the pharmacists in the study? Perhaps they graduated before 2014?

Line 306: Some schools are adding hormonal contraception experiences to IPPEs too.

Author Response

Thank you for the thoughtful review and responses. We have listed our responses below.

General Comments

This is an extremely well written and organized paper. The authors present a very interesting educational study about students’ confidence and abilities in providing hormonal contraceptive services after going through a curricular training. This paper adds to the existing literature. Response: Thank you!

Throughout the paper, indicating the training is for “prescribing” such as Line 245 is misleading. This may be a terminology difference, but what we are really training students and pharmacist is to provide hormonal contraception services, from which a prescription may result; a key component, as authors identify is the Plan step (actually making a decision on which hormonal contraception to choose/recommend). Would be helpful to add some language in the paper that students are being prepped to provide HC services (not just prescribing a drug). Response: this is a great point, and we went through and updated many of those instances to define it as a service, not just prescribing.

Specific Comments

Line 64: Do authors mean pharmacy students or pharmacists (as written)? Please clarify. Response: this paragraph was rearranged/reworded to better clarify that we do not know the answer for either students or pharmacists, but that looking at student learners could theoretically help us prepare graduates for these future roles.

Line 68: Suggest rewording that two examples of schools…As worded, makes it sound like only 2 schools have this content. Response: Updated

Line 86: More detail on what the “module” consists of would be helpful. In Line 91, authors indicate “lecture-based” material—I assume this is the module? How long is the lecture-based material? Was there any assigned reading or other material to review as self-study? Response: this section was clarified to explain that the modules are basically a lecture/series of lectures to deliver the content, plus student self-study.

Line 91: It isn’t clear what was new in the training, if anything. Was the pre-recorded 20 minute PPT a new component to the training? The authors’ hypothesis is that the current length of curricular content may be inadequate, which initially led me to think the study was going to assess 2 different types of training, but this isn’t the case. It appears that the addition of the simulation component was a new piece. Response: Updated some of the language in this paragraph to clarify which elements were new. The additional element was the prescribing simulation activity which served as both a measure of baseline ability and an intervention: this study was trying to determine if students were able to follow the prescribing process and come to a safe and effective recommendation after having “normal” training - if clearly not, then that may indicate the need for updated curriculum and/or cement the need for additional pharmacist training post-graduation in order to offer theses services.

Line 109: Suggest use of term “seeking” instead of “looking”. Response: Updated

Line 116: How large were the student groups? Did the interview with the SP occur in a Simulation Center? Or other type of room? Please clarify. Response: Updated with this information.

Lines 152-153: Where is this data presented? How many students indicated 4-6 hours? Suggest including the mean/median of self-study hours (as this is also cited on Line 173). Response: added an additional table with this information to the demographic results section.

Lines 184-185: I am confused why the %’s are negative if confidence increased post-training/simulation. Please clarify. Response: those were not intended to be negative numbers, they were dashes to represent relationship to the PPCP steps. Updated for clarity.

Line 265: Not quite sure the actual implications here. What year were the pharmacists in the study? Perhaps they graduated before 2014? Response: Updated to clarify this further.

Line 306: Some schools are adding hormonal contraception experiences to IPPEs too. Response: Noted and updated.

Reviewer 2 Report

Overall Comments:

Very well written, and loved the topic.  Recommend this for publication with only minor edits, although none on methodology.

Suggested Edits:

Title: I get that it will make the title longer, but may need to spell out PPCP. Non-pharmacists likely have no idea what that is, and could be confusing for them.

Abstract:  First 2 sentences don't quite make the correlation between states permitting pharmacists to prescribe contraception and the aim of the study.  Suggest adding in a short sentence stating that because of the expanded scope of practice, the colleges of pharmacy's in the study have incorporated this into their curriculum.

Line 43-44: Would recommend removing the sentence about legislation that has been introduced to recognize pharmacists as providers.  While I agree that it has some place in the discussion of pharmacist prescribing, given that it hasn't happened yet, don't know that it contributes much to this topic without going into much more detail.

Lines 51-54: This is probably nit-picky, but the referenced study here only provided the data on the number of prevented unintended pregnancies and the cost savings because of it.  The 10% of new prescriptions came from a different study that the cost-savings study used data from.  They were both published at the same time, so would recommend adding the other one.

Lines 68-73: Since these are the schools in this study, recommend moving this paragraph down into the methods section.

Line 95: Not necessary, but may be helpful for other colleges of pharmacy to know which state protocol was used as most every state has their own.

Lines 164-166: Suggest reiterating that this is data from prior to the simulation exercise.

Lines 285-286: Great point here about self-perceived confidence levels not always matching up to performance.

Line 334: Like that you were able to suggest a number of hours dedicated to this topic in a pharmacy school curriculum.

Author Response

Thank you for your thoughtful review and response! Our responses are listed below.

Overall Comments:

Very well written, and loved the topic.  Recommend this for publication with only minor edits, although none on methodology.

Suggested Edits:

Title: I get that it will make the title longer, but may need to spell out PPCP. Non-pharmacists likely have no idea what that is, and could be confusing for them. Response: Updated

Abstract:  First 2 sentences don't quite make the correlation between states permitting pharmacists to prescribe contraception and the aim of the study.  Suggest adding in a short sentence stating that because of the expanded scope of practice, the colleges of pharmacy's in the study have incorporated this into their curriculum. Response: Very helpful suggestion, updated as suggested.

Line 43-44: Would recommend removing the sentence about legislation that has been introduced to recognize pharmacists as providers.  While I agree that it has some place in the discussion of pharmacist prescribing, given that it hasn't happened yet, don't know that it contributes much to this topic without going into much more detail. Response: Thank you for the suggestion, we want to make sure it is mentioned, but attempted to make a more clear connection about its presence in the paragraph.

Lines 51-54: This is probably nit-picky, but the referenced study here only provided the data on the number of prevented unintended pregnancies and the cost savings because of it.  The 10% of new prescriptions came from a different study that the cost-savings study used data from.  They were both published at the same time, so would recommend adding the other one. Response: Thank you for pointing this out, we have clarified that sentence.

Lines 68-73: Since these are the schools in this study, recommend moving this paragraph down into the methods section. Response: updated

Line 95: Not necessary, but may be helpful for other colleges of pharmacy to know which state protocol was used as most every state has their own. Response: Updated to specify Oregon. Of note, the protocol the students followed was a hybrid of Oregon and California (which offered greater pharmacist authority and an expanded formulary). This information is now noted later in that paragraph.

Lines 164-166: Suggest reiterating that this is data from prior to the simulation exercise. Response: Updated

Lines 285-286: Great point here about self-perceived confidence levels not always matching up to performance.

Line 334: Like that you were able to suggest a number of hours dedicated to this topic in a pharmacy school curriculum.